# Lessons from the evaluation of the South African National Female Condom Programme

Mags Beksinska[1]*, Phumla Nkosi[1], Zonke Mabude[1], Joanne E. Mantell[2], Bongiwe Zulu[1], Cecilia Milford[1], Jennifer A. Smit[1]

1 MatCH Research Unit (MRU), Department of Obstetrics and Gynaecology, Faculty of Health Sciences, University of the Witwatersrand, Johannesburg, South Africa, 2 Division of Gender, Department of Psychiatry, HIV Center for Clinical and Behavioral Studies, Sexuality and Health, New York State Psychiatric Institute and Columbia University Irving Medical Center, New York, New York, United States of America

* mbeksinska@mru.ac.za

## Abstract

### Background

Understanding of the facilitators and challenges to female condom (FC) uptake has been limited due to lack of evaluation of national FC programmes.

### Setting

The FC has been an integral component of South Africa's (SA) HIV prevention programme for 20 years and is the largest government-funded FC programme worldwide.

### Methods

The national FC evaluation used a mixed-methods approach and consisted of key informant interviews and a telephone survey in a national sample of public and non-public sites. A sub-sample of sites participated in client and provider interviews, and a self-administered client survey. A review of distribution statistics from South Africa's District Health Information System was also conducted.

### Results

All 256 public-sector and 28 non-public-sector facilities reported having ever distributed FCs. Less than 5% of these facilities reported stock-outs and less than 3% reported they had a supply of expired female condoms. Systems for male condom (MC) and FC distribution were complementary, with similar ordering, delivery and reporting processes. FC promotion by providers (n = 278) varied with regard to FC training, whether attitudes about FCs influenced providers offer of FCs, and how they counselled clients about FCs. Of the 4442 self-administered client surveys in 133 facilities, similar proportions of women (15.4%) and men (15.2%) had ever used FCs. Although FCs were available at almost all sites surveyed, only two-thirds of clients were aware of their availability.

**Data Availability Statement:** All relevant data are within the manuscript and its Supporting Information files.

**Funding:** This study was funded by the United States Agency for International Development

(USAID) under AID-OAA-A-13-00069 (Mags Beksinska, PhD, and Jennifer A. Smit, PhD, Co-Principal Investigators). Joanne Mantell, MS, MSPH, PhD, was also supported by a center grant from the National Institute of Mental Health (P30-MH43520; Principal Investigator: Robert H Remien, PhD. The content of this article is solely the responsibility of the authors and does not necessarily represent the official views of USAID, and the HIV Center for Clinical and Behavioral Studies at the New York State Psychiatric Institute and Columbia University, Department of Psychiatry.

**Competing interests:** The authors have declared that no competing interests exist.

## Conclusion

Data highlight the role of providers as gatekeepers to FC access in public and non-public sectors and provide support for further FC programme expansion in SA and globally.

## Introduction

The female condom (FC) is an important multipurpose prevention technology (MPT) combining protection against unintended pregnancy, HIV and sexually transmitted infections (STIs) [1], and is the only female-initiated HIV prevention barrier method. Female condoms protect against pregnancy 95% of the time during perfect use, and 79% of the time during typical use [2]. The polyurethane FC1 has demonstrated high efficacy against HIV and STIs, but there are no published studies on the efficacy of newer FC designs in preventing HIV and STIs [3].

Lack of commitment by major donors to support FC programming has meant that FCs have not been an accessible prevention option in many of the countries hardest hit by HIV and unintended pregnancy [4]. A comprehensive analysis of why the female condom has not reached its full potential is attributed to a lack of acceptability in the international policy arena, which has led to a reticence to support its introduction, rather than grounded in potential users' perspectives [4]. The FC was identified by the Reproductive Health Supplies Coalition as one of several under-used reproductive health technologies having the potential to expand choice in reproductive health and family planning programmes [5]. However, despite increased FC distribution globally [6], distribution remains low relative to male condoms (MCs), accounting for only 0.19% of global condom procurement [5], and this imbalance is reflected in donor commitment [6–8]. Despite this, there has been significant progress in FC technology, with three new FC brands prequalified by the World Health Organization (WHO)/United Nations Population Fund (UNFPA) since 2012 [9], and others are under development [10].

The South African government launched a formal, three-phase "Female Condom Introduction Programme" in 1998 [11, 12] which focused initially on family planning clinics, promoting the FC as a dual method for preventing pregnancy and disease. With the programme's geographical expansion, the government complemented the public-sector programme with the donation of free FCs on request to non-governmental organisations (NGOs). In the context of extremely high HIV and unintended pregnancy rates, the South African programme has been scaled up considerably. In 2012, South Africa procured one billion MCs and 11 million FCs, with an aim to ensure the availability of at least one FC distribution site in all of the 254 sub-districts in the country [13]. By 2014 the National Department of Health (NDoH) made FCs available to all public-sector sites, expanded distribution to non-public sites, tertiary institutions and added two new FC products (Cupid and Pleasuremore), thus increasing consumer choices of barrier methods. While MCs had been socially marketed in South Africa for 20 years [14], FCs were not added to the socially marketed "Lovers Plus" brand until 2015. The Lovers Plus FC (FC2) was rebranded as an "inner condom" using the same Lovers Plus packaging for male condoms for public sector distribution. The rebranding stemmed from the concern that marketing the product as the 'female' condom may limit appeal to potential male purchasers. However, the socially marketed FC was discontinued a year later due to poor sales.

Distribution targets for FCs and MCs have been set in South Africa's National Strategic Plans [15, 16]; 25 million FCs were to be distributed yearly by 2016 [15], a goal that was

exceeded by 2 million. FCs were available for distribution not only in health facilities but non-traditional venues such as airports, hotels, shebeens (bars), tertiary institutions, mines and correctional facilities. In the 2017–2022 Strategic Plan [16] a target of 40 million FCs was set for FY 2021–2022. Despite these distribution targets, FC uptake has been low–only 7.2% in a population-level survey conducted in 2008 [17], and a recent review of the FC in South Africa found that FC use ranged between 2.9% and 38.7% [18]. Today, South Africa has one of the most robust FC programmes globally.

There is a dearth of comprehensive evaluation data on national FC programmes globally. South Africa, Brazil and India have the largest FC programmes supported by ministries of health, yet no large-scale comprehensive evaluation of these countries' programmes has been conducted [8]. This lack of information limits our understanding of factors related to FC distribution, commodity availability at service delivery points, uptake and use–and leaves many policy, programmatic and user questions unresolved. To address this gap, we conducted a process evaluation of the South Africa's National Female Condom Programme to understand what and how FC programming is being implemented and programme strengths and weaknesses so as to provide guidance to the South African government on how to improve FC access and uptake.

## Materials and methods

### Selection of sites

We selected both public sector and non-public sector sites. The public health sector sampling frame comprised the *National STIs Sentinel Surveillance Sites* which include approximately 30 sites in each of the nine provinces (n = 270) [19]. The non-public sector sites aimed to include one non-governmental organization (NGO), one tertiary education institution, one social-marketing outlet, and one private-sector site in each province (n = 36). Tertiary institutions were selected because they are sites for national female and male programming and social marketing outlets were targeted due to the South African government's launch of a socially marketed FC managed by the Society of Family Health (local affiliate of Population Services International). Social marketing outlets refer to retail channels such as stores and petrol stations. Sites were randomly selected where possible from a list of non-public FC-distributing sites. All public and non-public sector sites were asked to participate in a telephone interview.

The on-site assessment sample was selected randomly and proportionally from the STI surveillance sites based on four criteria: (1) location (rural, urban, peri-urban); (2) level of care (community health center, primary health care (PHC) clinic; (3) well-established long-term FC distribution (>5 years) and newer sites (<2 years); and (4) sites distributing different brands of FC products. Between 11–14 sites were selected per province.

### Sampling and data collection methods

The national FC evaluation used a convergent parallel design based on a combination of seven discrete data collection methods, primarily quantitative, from diverse groups of participants to obtain a comprehensive understanding of the context of service-delivery challenges, from commodity procurement, distribution and storage to availability and uptake via triangulation to validate findings from these sources (see Table 1) [20]. Qualitative and quantitative data were collected simultaneously, although with some time lags between data collection, and were analysed separately, and then integrated in final analysis [20]. The first level of data collection consisted of key informant interviews, a desk review of national FC distribution statistics, and a telephone survey of public and non-public sector sites that offered the female condom. In the second level, we drew a sub-sample of these public and non-public sector sites to participate in

**Table 1. National female condom evaluation data sources, data type, target populations and sites.**

| Data Source | Data Type | Domains for This Analysis | Target Population and Target Numbers | Site |
|---|---|---|---|---|
| **Key Informant Interview** | Qualitative | National government policies and programmes, procurement, condom distribution, storage, supply chain management, availability in provincial and district service delivery sites, monitoring & evaluation, demand creation for FCs | 20–25 policymakers, programme managers, individuals involved in condom social marketing strategies at a district, provincial and national level | Country-wide; not site-specific |
| **Telephonic Survey** | Quantitative | Condom procurement, stock-outs, storage and distribution | 270 public sector sentinel surveillance sites. | All sites |
| | | DHIS condom distribution data (compare with 3-month data given on telephonic survey | 36 non-public sector sites—one of each of following categories per province: 1. Tertiary education, 2. NGO, 3. Private, 4. Social marketing, 270 public sector sites* | |
| **Site Assessment** | Quantitative | Condom distribution, availability of female vs. male condoms, IEC materials and condom models, health education talks, knowledge and attitudes about condoms | Up to 150 sites | Sub-set of sites |
| **Provider Interview** | Quantitative | FC and MC training and perceived need for more training, whether provider discussed condom use with female clients, whether provider gave condoms to male and female clients, whether provider demonstrated FC use to new users and reasons for not always demonstrating use, frequency of FC and MC education of clients, and FC attitudes. | 278 providers | Sub-set of sites |
| **Client Exit Interview** | Quantitative | First FC use, source of first FC, reasons for use, whether provider explained how to use FC and adequacy of information, preference for type of condom, whether offered FC by provider or requested by client | 427 female clients, 18–49 years, who were current or ex-users of FC | Sub-set of sites |
| **Client Anonymous Survey** | Quantitative | FC awareness, knew of FC availability at site, whether offered FC by provider, ever used FC, and if so, whether used with partner, condom preference, reasons never tried using FC | 4442 female and male clients | Sub-set of sites |

*Non-public sector sites did not report to DHIS at time of study.

site assessments, provider interviews, client anonymous surveys, and client exit interviews. All data collection was conducted by a cadre of research interviewers trained in quantitative and qualitative methods. Two sub-studies–a descriptive cost study and a cohort of women who were new FC users, and among a sub-set of these women, their male partners–under the FC evaluation initiative were conducted in only one province and are outside the scope of the national evaluation.

Because SA has an integrated female and male condom programme, data on MCs were collected in all components of the evaluation, although in less detail.

The client anonymous survey and client exit interview as well as consent form for the latter were translated into all of the 11 South African languages and back-translated by experienced translators to check for accuracy. The client interviews and consents were conducted in participants' language of choice.

**Key informant interviews.** Key informant interviews were qualitative in nature and designed to elicit a better understanding of the context of system-level FC issues. We purposively selected policymakers and programme managers as well as individuals involved in social marketing strategies at a district, provincial and national level to ensure representation of a range of views about the FC for qualitative key informant interviews. An initial list of key informants (KIs) was drawn up based on the research team's knowledge of key role players in the public and non-public healthcare sectors) and discussion with the National Department of Health. Criteria for eligibility was a minimum of one year in current position and we aimed to conduct between 20 and 25 interviews, depending on data saturation and availability of key

informants. Interviews were conducted either face-to-face or via telephone. The interviews aimed to identify critical issues in the FC chain, including advocacy, overall programme leadership and coordination, supply and commodity security, provider training, monitoring, and integration with other programmes. Audio-recordings were transcribed.

**Review of condom distribution statistics from the District Health Information System (DHIS).** The District Health Information System (DHIS) is a web-based data analytics and information system that tracks health service delivery in the public health sector for South Africa. DHIS data are used for health service planning and monitoring. In this evaluation, we reviewed DHIS condom distribution data for each participating site for the same three months that site assessment data were inspected (February-April 2014).

**Telephonic surveys of public and non-public sector sites.** In the telephonic survey we collected information over the telephone by conducting an interview with the facility manager or their designee. The questionnaire and consent form was sent in advance to allow time for the data required to be collated. The survey included questions on numbers of male and female condoms distributed, where dispensing to clients takes place, distribution to other sites, availability of Information, Education, and Communication (IEC) materials and condom models, constellation of staff, and staff training on the FC. This information was subsequently verified in the on-site assessment.

**Female condom-distributing site assessments.** The site assessments were scheduled following completion of the telephonic interview. The quantitative site assessments focused on condom storage, supply and distribution issues (e.g., distribution points, stock-outs, expired stock, sub-distribution to other sites); availability of IEC materials and condom demonstration models; and provider condom education talks to clients.

**Provider interviews.** We purposively selected a sample of providers, including operational managers and clinicians, for quantitative interviews. The number of interviews per site was based on the total number of staff employed at the facility. If the total staff number was five or more, we interviewed three staff; if less than five, two staff were interviewed. The quantitative interview focused on female condom knowledge and attitudes, counselling practices, and condom dispensing and logistics training. We included 24 items assessing attitudes about the FC, e.g., sexual pleasure, inconvenience, improved prophylaxis, insertion reluctance, and trust [21].

**Client anonymous surveys.** These anonymous self-administered brief quantitative surveys were completed by women and men and were designed to be completed in one to two minutes. On the morning of the site assessment, research staff introduced the study in waiting areas and invited all clients in this area to participate. In addition, surveys were left at reception as well as on tables and unoccupied chairs. Participants were asked about their awareness of the FC, whether ever offered the FC, and if so, whether they accepted this offer, ever used the FC, and if used, reasons for first use, where they obtained FCs, the counselling received, and their experience using the product. Never FC users were asked about their reasons for non-use.

**Client exit interviews.** These quantitative interviews were conducted with women aged 18–49. Data from women who participated in these interviews was not linked with data from women who participated in the anonymous surveys. During the visit to the facility, research staff informed clients during the day in the different waiting areas that any ever or current users could participate in an interview at the end of their consultation. The interview duration was about one hour. Data presented here focused on reasons for first FC use, source of receiving first FC, whether FCs were offered by providers or requested by the client, whether provider explained how to use the FC and adequacy of information, and if offered, choice of condom type.

## Data analysis

Quantitative data were captured and descriptively analyzed using StataIC v14 (StataCorp, College Station, TX). Pearson's chi-square or Fisher's exact test of association were calculated for categorical variables.

Thematic analysis of the key informant interview data was conducted, identifying important and emerging themes. An initial codebook was created collaboratively by team members; and a portion of transcripts were double-coded to test the codebook. Discrepancies in coding and application of codes to the transcripts were resolved in team meetings of coders so as to improve inter-rater coding reliability. The remaining transcripts were also double-coded. The final codebook reflected codes developed both inductively and deductively. NVivo (Version 10, QSR International) was used to facilitate systematic data management.

A unit-cost analysis was conducted at eight sites in order to establish FC program costs; findings from this analysis are reported elsewhere [22].

## Ethical considerations

The evaluation was approved by the Human Research Ethics Committee (HREC) at the University of the Witwatersrand (M140428/M140365). Written informed consent was obtained from all participants (except for the self-administered client anonymous survey).

## Results

### Description of sample

Data were collected between August 2014 and September 2016. Of the 270 sentinel surveillance sites, six declined to participate, and a further eight were excluded due to boundary changes, mergers and closures, resulting in 256 public-sector sites. Twenty-eight non-public-sector sites were included. Although an NGO distributing FCs was identified in each province, not all provinces had all three categories (tertiary education, social marketing, and private sector).

Table 2 presents the sample size for each data collection method by province. The number of telephone surveys and site assessments conducted was similar across the nine provinces. Variation was due to the 14 sites excluded as explained in the previous paragraph and the availability of non-public sector sites in each province. Some of the smaller clinic in rural areas such as the Eastern Cape had few clients that had ever used FCs available for interview.

Table 2. Sample size by data collection method and province.

| Province | Phone Survey N = 284 | Site Assessment N = 133 | Provider Interview N = 278 | Client Exit Interview N = 426 | Anonymous Survey N = 4442 | Key Informant Interviews N = 26 |
|---|---|---|---|---|---|---|
| Eastern Cape | 30 | 14 | 27 | 11 | 149 | 3 |
| Free State | 31 | 13 | 20 | 52 | 436 | 1 |
| Gauteng | 32 | 16 | 30 | 83 | 780 | 1 |
| KwaZulu-Natal | 34 | 15 | 34 | 63 | 575 | 4 |
| Limpopo | 33 | 16 | 47 | 59 | 508 | 1 |
| Mpumalanga | 31 | 16 | 26 | 46 | 504 | 1 |
| Northern Cape | 31 | 15 | 34 | 24 | 369 | 1 |
| North West | 32 | 14 | 35 | 36 | 359 | 1 |
| Western Cape | 30 | 14 | 25 | 55 | 762 | 1 |
| National | - | - | - | - | - | 12 |

**Table 3. Distribution of sample of in-depth site assessments in each province by site location, facility type, years of FC distribution, and having more than one FC during site visit.**

| Province and Number of In-Depth Site Assessment Visits | Site Location | | | Facility Type | | | Years of FC Distribution | | | Sites with >1 FC* |
|---|---|---|---|---|---|---|---|---|---|---|
| | Rural | Peri-urban | Urban | PHC | CHC | Non-public | <2 | 2–5 | >5 | |
| Eastern Cape (n = 14) | 11 | 2 | 1 | 12 | 1 | 1 | 3 | 1 | 10 | 0 |
| Free State (n = 13) | 7 | 3 | 3 | 11 | 0 | 2 | 7 | 0 | 6 | 5 |
| Gauteng (n = 16) | 0 | 6 | 10 | 10 | 3 | 3 | 2 | 2 | 12 | 12 |
| KwaZulu-Natal (n = 15) | 8 | 3 | 4 | 11 | 1 | 3 | 3 | 1 | 11 | 1 |
| Limpopo (n = 16) | 11 | 2 | 3 | 12 | 1 | 3 | 5 | 0 | 11 | 11 |
| Mpumalanga (n = 16) | 13 | 1 | 2 | 12 | 2 | 2 | 3 | 2 | 11 | 1 |
| Northern Cape (n = 15) | 9 | 4 | 2 | 11 | 2 | 2 | 2 | 1 | 12 | 13 |
| North West (n = 14) | 8 | 3 | 3 | 10 | 1 | 3 | 2 | 0 | 12 | 7 |
| Western Cape (n = 14) | 7 | 1 | 6 | 10 | 2 | 2 | 2 | 1 | 11 | 13 |

*Refers to site having >1 FC type (brand) on site.

PHC = Public Sector Primary Health Clinic.

CHC = Public Sector Community Health Center.

Limpopo had the largest number of providers interviewed and this was because all sites aside from one had a large staff compliment and staff were made available for interview.

Table 3 presents the distribution of the sample for in-depth site assessments in each province according to site location (rural, peri-urban, or urban), facility type (public sector primary health clinic, public sector community health center, or non-public sector site), length of FC distribution, and number of sites distributing more than one type of FC. Provinces differed by proportion of site location (rural/peri-urban/urban) according to the proportion of sites of a similar location in the National Surveillance Site sample. There were fewer Community Health Centres in the National sample, with some rural provinces not having a CHC in the sample or only one. The length of FC distribution was found to be greater than five years in most sites, with some sites commencing the programme in the last few years and fewer in the 2- to 5-year range of distribution. This distribution also reflected the National FC Programme which was implemented in a phased approach. The aim was to include all non-public sector sites participating in the telephone survey in the on-site assessment; however, only 19 of the total 28 participated for a number of reasons. Some private sites were unable to accommodate the research team during the time of the provincial visit or the FC programme was delivered in a way that potential clients were not available for interview on site (condoms delivered to community directly).

## Themes

Results are categorized into four sections, starting with system issues and then shifting to health care provider and client/user issues: (1) Overview of national government policies and programmes; (2) Commodity procurement, supply chain management. distribution, availability, and monitoring and evaluation in provincial and district service delivery venues; (3) Provider experience in the female condom programme; and (4) Client experiences with the FC. Comparable data are synthesized across data collection methods.

## Overview of national government policies and programmes

Key informants provided their perspectives on the national FC programme and many discussed condoms in general, rather than differentiating between FC and MC. They noted that

FC and MC programming is integrated in health settings, educational institutions and work-places. Despite this integration, key informants indicated a lack of harmonisation of policies, especially across health, education and social security sectors.

The condom programme is funded primarily by the national SA government, with some additional support from international funders. Despite annual increases in FC budgets, the programme was thought to be challenged by insufficient funding, largely because FC demand is higher than allocated budgets. FC and MC programming policies are integrated. There are varying degrees of national involvement in developing national policy for condom program-ming, some of which is done collaboratively with partners such as PEPFAR, USAID and UNFPA. The National Department of Health also conducts policy reviews, disseminated poli-cies and monitored implementation of these policies.

Technical task teams, which include provincial and NGO representatives, have been estab-lished to review existing policies, especially regarding issues of access and sustainability, and set programming and research agendas. Condom programming policies are integrated into diverse health programmes, e.g., family planning, voluntary medical male circumcision, HIV counselling and testing as well as in other sectors, e.g., Department of Education. Research data from end users are used to inform policymaking about condom programming policies.

Condom distribution targets are set at national level, and these are divided into provincial and district targets. Population-distribution statistics and logistics-management systems are used to determine quantities of condoms. The National Condom Distribution Plan 2013–2016 provides managers with guidance on the implementation and evaluation of condom distribu-tion in their areas [23]. Monitoring and evaluation (M&E) systems are well-established, with FC distribution added to the DHIS reporting requirements in 2013.

### Commodity procurement, storage, supply chain management. distribution, availability, and monitoring and evaluation in provincial and district service delivery venues

The Logistics Management Information System, along with DHIS and M&E data, are used to estimate condom procurement and distribution needs, but key informants still identified chal-lenges. Procurement and storage systems for FCs and MCs were noted to be similar in all pub-lic-sector sites. Both types of condoms were collected or delivered within a week of ordering by 74.6% of sites, except for a few locations in rural areas, where it took up to six months. Three-quarters (77.8%) of sites had a store-room/dispensary for FCs and MCs, but a third (31.6%) noted that correct storage was a challenge, primarily because the condom boxes were kept directly on damp ground rather than in a dry environment without sunlight. Also, key informants indicated that storage capacity was limited.

Key informants noted various issues with the supply chain management process led to unavailability of stock (either complete absence or limited availability). These included delays from international manufacturers, inability of suppliers to meet demand for condoms, suppli-ers' unwillingness to register on the procurement database, monitoring condom commodities (e.g., exposure to sun, quality) from suppliers, costs of condoms due to fluctuations in exchange rates, expiration of supplier contracts, delayed payment to suppliers, withdrawal of suppliers from the contract after the award had been made. The majority of key informant indicated they had experienced FC stock-outs.

Telephone survey and site assessment data indicated that all 284 sites had ever distributed FCs, half (53.8%) for more than five years, and 18.7% commenced distribution within the last two years. Only a small proportion of sites (2.8%) had stock expire in the last year. Fourteen (4.9%) sites reported stock-outs for a range of reasons. These included a depleted FC supply

(n = 7), late ordering of FCs (n = 2), no demand for FCs, rumours that FCs were not being used for what they were intended, so staff did not re-order (n = 2); and one site identified itself as a non-designated FC distribution site.

FC distribution between sites in one three-month period during the evaluation varied widely—from no units to more than 200,000 units per month. In nearly three-fifths (57%) of the 114 public-sector sites participating in the on-site assessment, there was no agreement among the three data sources (telephone survey, site visit and DHIS) in at least one of the three months. Reasons for the discrepancies were mainly unknown or were assumed to be due to missing records.

Overall, a quarter (25.3%) of public-sector sites reported sub-distribution to other sites (NGOs, garages, taverns, brothels and taxi ranks) compared to 66% of NGOs that distributed FCs directly from their sites. Condom distribution in non-traditional outlets was seen as a way to increase access to FCs and decrease the possibility of stigma linked to accessing them in healthcare settings; however, often there was no trained person to support clients' FC use in these outlets.

*For the female condom, it's a little bit different could because you can't just put a you know, a box of female condoms down a shebeen. [. . .] we've gotten instructions from NDOH to not do that.*

(NGO-Gauteng Province)

Key informants also described partnerships between DoH and NGOs, where NGOs also provided distribution support, especially distributing condoms to hard-to-reach areas in rural communities as well as to key populations such as sex workers, men who have sex with men and youth.

Availability of the MC to clients was higher than that of the FC at all site distribution points aside from female toilets where similar percentages were reported (Table 4). FC leaflet availability was higher than that of MC, as FC instruction leaflets are provided by manufacturers of each brand, though limited to English. Availability of FC and MC condom IEC materials was similar but availability of demonstration models for MCs was higher than that for FC.

**Table 4. Condom, IEC materials and demonstration model availability by distribution area.**

|  | FC, N = 284 | MC, N = 284 |  |
|---|---|---|---|
| **Distribution points** | % N | % N | p-value |
| At least one consulting room | 79.2 (225) | 86.6 (246) | 0.026 |
| Waiting area | 65.1 (185) | 76.4 (217) | 0.004 |
| Female Toilet | 34.5 (93) | 33.8 (96) | 0.86 |
| Male toilet | 23.8 (64) | 40.5 (115) | <0.001 |
| Corridor | 9.8 (28) | 11.6 (33) | 0.59 |
| Outside site (wall/gate) | 19.0 (54) | 29.9 (85) | 0.003 |
| **IEC (leaflets/posters)** |  |  |  |
| Waiting area-leaflets | 29.9 (85) | 10.9 (31) | <0.001 |
| Consultation room-leaflets | 27.1 (77) | 13.0 (37) | <0.001 |
| Waiting room posters | 27.1 (77) | 13.7 (39) | <0.001 |
| Consultation room posters | 10.9 (31) | 3.9 (11) | 0.002 |
| **Demonstration model (dildo or vaginal model)** |  |  |  |
| Waiting area | 13.3 (38) | 40.1 (114) | <0.001 |
| Consultation rooms | 22.2 (63) | 77.8 (221) | <0.001 |

Key informants at all levels described the process of M&E which begins when condoms are procured, continues when condoms are delivered to primary distribution sites (and from primary to secondary sites), and includes the reporting stage. M&E of female and male condoms was reported to face several shortcomings—incorrect reporting of distribution data, missing deadlines for reporting, lack of dedicated condom logistics staff at district level, lack of reporting systems for non-public sector sites.

## Demand creation for female condoms

Key informants described that communications strategies for FCs were being developed, and that some of these were integrated into HIV prevention and reproductive health campaigns as well as campaigns for TB prevention, breastfeeding, and voluntary medical male circumcision. Awareness campaigns and road shows were the most frequently note methods of promoting condoms in the communities. Often health calendar events, e.g., National STI week and Women's Day, local electronic and print media, booths set up in shopping centres, street-based marketing, as well as door-to-door at people's homes, tertiary institutions and workplaces are used to promote FCs, with NGOs playing a vital role in this initiative.

Key informants suggested a number of other strategies to improve marketing of the FC: (1) more advocacy; (2) use of social media such as Facebook and Twitter; (3) tailoring strategies for rural and urban populations; and (4) targeting key populations, partners and older people who influence decision-making.

*It's where the aunties [are] so, I know that sometimes we spend time targeting the end user. We should be targeting the influencer- [. . .] -to the same extent that we target the end user, and also the males themselves.*

(NGO-Gauteng Province)

Harnessing the power of popular high-profile influencers was also seen as a potentially effective strategy for promoting FCs.

*It needs to start from up, right on top in, in, in government you know, filtering down to maybe people that, that identify with certain uhm, maybe actors or you know, celebrities, sport celebrities that can promote female condoms.*

(District DoH KwaZulu-Natal Province)

Messaging centred on pleasure, women and empowerment, rather than on HIV prevention which often is stigmatised, was seen as fundamental. Many key informants described the role of men as dominant and decision-makers due to culture, tradition and women's economic dependence on men, although this is changing.

*But the main issue I think it's uh, social and also cultural. Social in terms of a lot of women don't work and it's so difficult for them to force their partners who are breadwinners to use condoms. [. . ..] And also in terms of cultural issues of the submissiveness [. . .] of women.*

(NDoH-Gauteng Province)

*[O] riginally they, men would never accept especially with, when somebody is married. [. . .] would never, what you call, your husband would never accept a married woman to ha, come with a condom and to, to say now you cannot sleep with me without a condom but at least*

*now I think things are coming, are becoming better because of the awarenesses that [...] are there.*

(Provincial DoH-Eastern Cape Province)

To improve female condom programming at the provincial and district level, key informants suggested the need for dedicated transport for condom distribution, availability of cheaper FCs, more staff allocated to the condom programme, and peer educators to distribute condoms and provide education about the FC.

*...constraints that limit our female condom programme. [...] current distribution technique and the fact that you know we don't have anybody responsible [...] solely for, for like the logistics management of condom distribution.*

(District DoH-KwaZulu-Natal Province)

## Provider experience in the female condom programme

Of the 278 providers interviewed, 75% were nurses, and the remainder were health promotors, peer educators, community health workers and a wide range of individuals in specialist roles including a pharmacy assistant, social worker, practice manager and student counsellor (Table 5).

Almost all (89%) of the providers were women. More providers were trained and counselled on MCs compared to FCs (Table 6). Ever trained was associated with having counselled female clients on the FC in the last month ($X^2 = 6.78$, p = 0.0009). The three most common reasons for not always demonstrating condom use (FC or MC) were that clients could "read the instructions", lack of time during the consultation and lack of demonstration models.

When the FC was available in every consultation room, half (51.2%) of providers indicated they had given FCs to clients in the past week, while less than a fifth (18.4%) only did so when

**Table 5. Provider job roles.**

| Job role | N = 278, N % |
|---|---|
| Registered nurse | 59.0 (164) |
| Enrolled nurse | 6.5 (18) |
| Enrolled assistant | 7.2 (20) |
| Lay counsellor | 13.7 (38) |
| Peer educator | 4.3 (12) |
| Health promoter | 2.9 (8) |
| Community Health Worker | 1.8 (5) |
| Project Coordinator | 1.4(4) |
| Volunteer | 0.4 (1) |
| Courtesy Manager | 0.4 (1) |
| HCT Mentor/condom champion | 0.4 (1) |
| Intervention Facilitator | 0.4 (1) |
| Pharmacist assistant | 0.4 (1) |
| Social worker / STI / HIV | 0.4 (1) |
| Practice Manager | 0.4 (1) |
| Student counsellor/Support officer | 0.4 (1) |
| Nursing Assistant | 0.4 (1) |

**Table 6. Provider training, counselling and distribution practices by condom type.**

|  | FC % (n = 278) | MC % (n = 278) |
|---|---|---|
| **Ever trained** | 65.5 (182) | 79.1 (220) |
| **Requires more training** | 82.0 (214) | 69.4 (186) |
| **Provider discussed condom use with female clients in one-to-one session in last month** |  |  |
| Never | 11.4 (31) | 3.7 (37) |
| Less than half | 11.4 (31) | 5.9 (16) |
| Half the time | 6.5 (18) | 3.7 (10) |
| More than half | 7.8 (21) | 7.0 (19) |
| Almost all | 39.5 (107) | 61.0(166) |
| Depended on the client* | 23.2 (63) | 18.8 (51) |
| **Provider personally gave condoms to clients in last week** |  |  |
| Female client | 47.4 (129) | 70.0 (191) |
| Male client | 29.6 (83) | 79.6 (218) |
| **Provider demonstrates FC use to new users** |  |  |
| Always | 73.0 (203) | 80.5 (224) |
| Sometimes | 12.6 (35) | 10.4 (29) |
| Not unless asked by client | 5.4 (15) | 4.7 (13) |
| Never | 8.9 (25) | 2.5 (7) |
| **Reason for not always demonstrating condoms** | **N = 73** | N = 49 |
| No demonstration model | 24.0 (18) | 10.2 (5) |
| No time to demonstrate | 20.0 (15) | 22.4 (11) |
| Client can read the instructions | 17.3 (13) | 40.8 (20) |
| Not trained | 14.6 (11) | 0 (0) |
| Don't normally counsel on condoms or refer | 13.3 (10) | 4.0 (2) |
| Clients know how to use/no need to tell them | 4.0 (3) | 22.4 (11) |
| Clients don't like them/not interested | 6.6 (5) | 0 (0) |
| Other | 10.7 (8) | 0 (0) |

*This was a verbatim response that may reflect providers' reluctance to discuss condoms with particular types of clients, e.g., older or married women.

FCs were available in some or none of the consultation rooms; this difference was significant ($X^2$ = 8.04, p = 0.005). Cadre of staff was the strongest correlate of FC distribution to men, with 43.7% of lay counsellors/peer educators reporting FC distribution in the last week compared with 25.9% of nurses ($X^2$ = 7.8, p = 0.005).

The frequency of educating clients about the MC and FC at the sites was similar because the talks usually included both condoms. The number of talks carried out per site varied considerably, with 10.2% of sites reporting no condom education in the last three months and 42.0% reporting at least daily condom education talks to clients.

As shown in Table 7, most providers held positive attitudes about the ability of FCs to provide better protection against both pregnancy (76.2%) and STIs/HIV (74.7%) compared to the MC. However, some providers expressed unfavourable attitudes toward the product itself, with nearly two-fifths (37.7%) strongly or somewhat agreeing that FCs were weird and some strongly or somewhat agreeing they were unappealing since part of the FC hung outside of the vagina (32.7%) or were messy (15.1%). Up to 25% of providers indicated "don't know" to many of the attitudinal statements.

Providers' technical knowledge of FCs differed. Most (84.5%) were aware that the FC should not be reused and of correct removal technique (84.5%); however, there was some confusion about lubricant use with FCs, with 19% believing that any type of lubricant could be used. Ninety percent of providers reported that clients were informed verbally about the availability of FCs and MCs, and most indicated a lack of availability of IEC materials. Almost all providers (96%) in sites that had experience with more than one FC product thought it was important to increase choice of FCs for clients.

## Client experiences with the FC

Of the 4442 anonymous client surveys, the majority were completed by women (87%). The age range was 18–57 years and the mean age of women (29.3 years, SD = 7.8) and men (29.9 years, SD = 8.2) was the same. Similar proportions of women (84.5%) and men (78.5%) had ever heard of FCs but fewer had ever used them (Table 8). There were no differences between male and female clients who reported ever FC use. However, among clients with current partners, women were about twice as likely to indicate they always used a FC with this partner. Ever use varied widely between the provinces (Fig 1) and was lowest (5.5%) in women under 20 years.

Table 8 shows client knowledge of FCs. Women were more likely than men to know that FCs were available at the clinic ($X^2$ = 19.6, p<0.001) and to have been offered a FC by a provider ($X^2$ = 35.1, p<0.001). Approximately half of female clients but fewer male clients had ever been offered a FC by a provider. Although FCs were in stock in almost all sites evaluated, not all clients were aware of this.

Men and women who had never tried using the FC gave the same main reasons, including using another contraceptive, being frightened to try it, and not knowing were to get them (Fig 2).

Four hundred and twenty-seven women, all of whom were current or ex-users of FCs, completed an exit interview. The mean age was 31.5 years (range 18–49, SD = 7.5 years), with only 2.8% under 20 years. Slightly more than two-fifths (42%) were HIV-positive and one-fifth (20.8%) reported having a STI in the last year. Main reasons for first use of FC, where FC first obtained and other experiences in the programme are shown in Table 9.

Of the current FC users, 73.4% reported using male condoms more often than before they started using FCs. Almost all (98.4%) of female clients reported that when they first received a FC from a provider, the provider had explained to how to use them. Most clients reported they were shown a FC, but only a third (32.8%) were given a demonstration of insertion on a model. Two-thirds of women (67.2%) received advice from providers on how to introduce the FC into their relationship. Almost all (96%, n = 51) of clients interviewed in sites which had more than one FC product available gave positive feedback regarding availability of a wider selection of FC products.

## Discussion

The SA FC programme is well-established and embedded in the healthcare system; systems for MC and FC exist for ordering, reporting, provider training and health education talks. In the early years of the SA programme, FCs were primarily distributed from provider consulting rooms to ensure that new users were given counselling on use and because of concerns about limited stock [13]. This mode of distribution is now shifting as FCs are also being offered in non-clinic settings.

Data highlight the role of providers as gatekeepers to FC access in public and non-public sectors; however, many clients were unaware of FC availability in their facilities. Providers held positive attitudes about the FC, e.g., believing that it provided better protection against

Table 7. Providers attitudes about the female condom.

| Item Description | Strongly Agree, N = 278, % N | Somewhat Agree, N = 278, % N | Somewhat Disagree, N = 278, % N | Strongly Disagree, N = 278, % N | Don't Know, N = 278, % N |
|---|---|---|---|---|---|
| Female condoms make sex better for women. | 51.8 (144) | 18.7 (52) | 6.5 (18) | 7.2 (20) | 15.8 (44) |
| Female condoms feel more natural than regular male condoms. | 47.1 (131) | 19.1 (53) | 10.4 (29) | 7.6 (21) | 15.8 (44) |
| Female condoms make sex last long. | 35.3 (98) | 18.7 (52) | 12.2 (34) | 8.6 (24) | 25.2 (70) |
| Female condoms are better than male condoms. | 47.8 (133) | 16.9 (47) | 11.5 (32) | 11.9 (33) | 11.9 (33) |
| Female condoms are weird. | 24.1 (67) | 13.7 (38) | 21.9 (61) | 36.3 (101) | 4.0 (11) |
| Female condoms are inconvenient. | 15.5 (43) | 12.6 (35) | 20.9 (58) | 48.2 (134) | 2.9 (8) |
| Female condoms are messy. | 10.1 (28) | 5.0 (14) | 20.5 (57) | 58.3 (162) | 6.1 (17) |
| Having part of the female condom outside the vagina is unappealing/turn-off. | 17.6 (49) | 15.1 (42) | 19.4 (54) | 41.4 (115) | 6.5 (18) |
| Female condoms offer better protection against unintended pregnancy than male condoms do. | 65.1 (181) | 11.6 (31) | 10.1 (28) | 11.5 (32) | 2.2 (6) |
| Female condoms offer better protection against sexually transmitted diseases than male condoms do. | 62.6 (174) | 12.2 (34) | 13.0 (36) | 1.04 (29) | 1.8 (5) |
| Female condoms are stronger than male condoms. | 65.1 (181) | 10.8 (30) | 6.1 (17) | 10.4 (29) | 7.6 (21) |
| The female condom takes too long to put in. | 20.5 (57) | 13.7 (38) | 19.8 (55) | 39.6 (110) | 6.5 (18) |
| It is hard to carry female condoms in a purse because of their size. | 15.8 (44) | 10.8 (30) | 18.4 (51) | 53.2 (148) | 1.8 (5) |
| Female condoms put the woman in charge. | 74.8 (208) | 10.8 (30) | 2.9 (8) | 8.3 (23) | 3.2 (9) |
| The female condom provides women another contraceptive choice. | 94.6 (263) | 3.6 (10) | 1.1 (3) | 0 (0) | 0.7 (2) |
| The female condom provides women another choice to protect themselves against HIV and other sexually transmitted diseases. | 97.8 (272) | 2.2 (6) | 0 (0) | 0 (0) | 0 (0) |
| Sex doesn't feel as good when you use a female condom. | 5.4 (15) | 5.8 (16) | 20.1 (56) | 48.6 (135) | 20.1 (56) |
| Female condoms make it hard for a woman to have an orgasm (cum). | 2.9 (8) | 3.6 (10) | 22.3 (62) | 49.6 (138) | 21.6 (60) |
| Female condoms make it hard for a man to have an orgasm (cum). | 1.4 (4) | 3.6 (10) | 23.7 (66) | 50.7 (141) | 20.5 (57) |
| Female condoms take all the fun out of sex. | 2.9 (8) | 5.4 (15) | 22.7 (63) | 58.3 (162) | 10.8 (30) |
| You don't like the idea or thought of putting the female condom inside yourself. | 13.3 (36) | 9.2 (25) | 17.3 (47) | 53.5 (145) | 6.6 (18) |
| You don't like the idea or thought of having to touch yourself to put the female condom in. | 13.3 (36) | 5.5 (15) | 19.6 (53) | 56.1 (152) | 5.5 (15) |
| You don't like the idea or thought of having to use your finger to put the female condom in. | 10.3 (28) | 6.6 (18) | 19.2 (52) | 58.7 (159) | 5.2 (14) |
| If a woman wants to use a female condom, her partner might think she was having sex with someone else. | 16.9 (47) | 16.2 (45) | 15.8 (44) | 47.8 (133) | 3.2 (9) |

unintended pregnancy and HIV/STIs than male condoms, was stronger than the male condom, and made sex better for women. However, they disliked some product features, e.g., its messiness and hanging outside of the vagina. It is not surprising that 15% of the providers were unaware that FCs could not be re-used. At the time of this evaluation, FC1, which was made of polyurethane, was replaced by FC2, which is made of synthetic latex. Research on the polyurethane FC1 indicated that it could be reused several times after washing and drying, but this was no longer possible with FC2 which is made of synthetic latex [24]. A small number of providers who participated in this evaluation may not have realized the difference between the two FCs in terms of re-use.

**Table 8. Clients' knowledge and usage of FC in South Africa.**

| Knowledge and use of FC | Women (n = 3821)[a] % n | Men (n = 577)[a] % n | p-value |
|---|---|---|---|
| Ever heard of FC | 84.5 (3148) | 78.5 (453) | 0.028 |
| Knew of FC availability at site[b] | 72.8 (2293) | 63.3 (287) | <0.001 |
| Ever offered by any provider[c] | 52.0 (1489) | 35.9 (139) | <0.001 |
| Ever used FC | 18.6 (587) | 19.4 (88) | 0.74 |
| If ever used, use with current partner[d] | N = 522 | N = 83 | 0.023 |
| Never | 10.5 (55) | 13.3 (11) | |
| Sometimes | 45.4 (237) | 54.2 (45) | |
| Often | 19.2 (100) | 10.8 (9) | |
| Always | 24.9 (130) | 12.0 (10) | |

[a] A small number (<1.0%) of the 4442 survey participants did not respond to the "male/female" question.

[b] Asked only of those who had ever heard of the FC.

[c] Asked only of those who had heard of FC and who knew it was available at the site.

[d] Asked only of those who responded that they had a current partner.

Provider training was linked to counselling and distribution and a high proportion of providers expressed a need for more training, perhaps linked to the increased availability of FCs at sites since 2014. A need for more provider FC training has been noted in other studies in the region [25, 26].

FC users under the age of 20 years were poorly represented in the FC evaluation in both the anonymous survey and client interviews, in agreement with national surveys where users in the 15–19 age group reported the lowest percentage of ever FC use [27–29]. This points to a need to focus on FC promotion to younger clients. Client reasons for not using FCs were similar across all data-collection methodologies. A high proportion of FC users were HIV-positive, indicating successful promotion to this group of women.

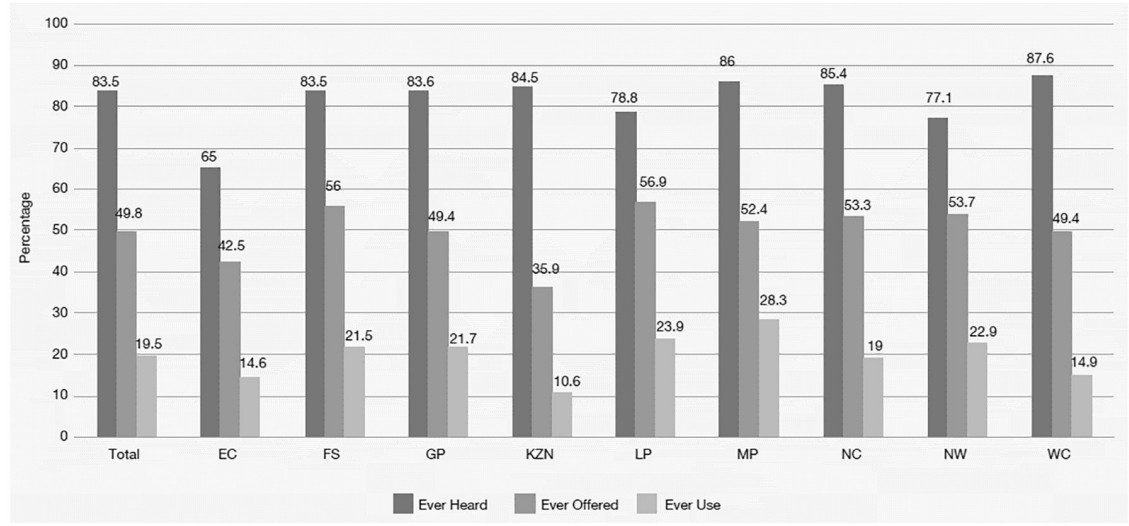

**Fig 1. 'Ever heard of' FC, 'ever offered' FC and 'ever used' FC, by province.** EC = Eastern Cape, FS = Free State, GP = Gauteng Province, LP = Limpopo, MP = Mpumalanga, NC = Northern Cape, MW = North West, WC = Western Cape.

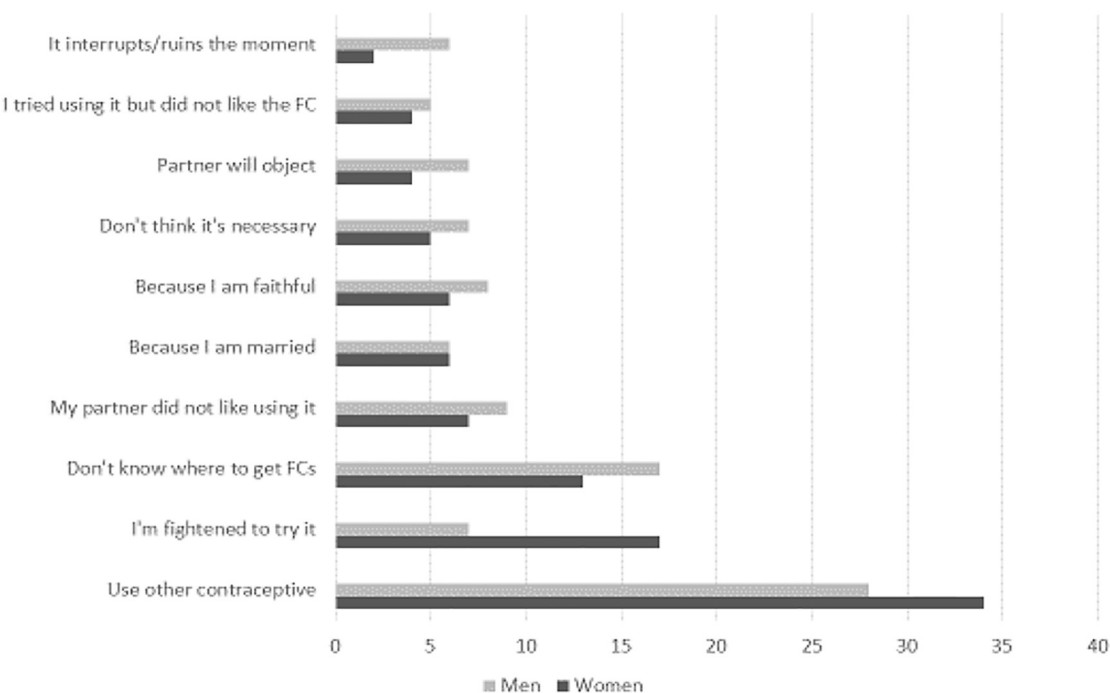

**Fig 2. Female and male clients' reasons for not using FCs.**

Table 9. Female clients' experiences with FC use.

| Experience | % (n = 427) |
|---|---|
| **Main reasons for first FC use** | |
| to protect against HIV/STIs | 39.4 (166) |
| to protect against pregnancy | 40.9 (172) |
| just wanted to try one | 28.0 (117) |
| **Source of obtaining first FC** | |
| Providers | 76.3 (326) |
| Condom dispenser | 17.8 (76) |
| Friend | 3.3 (14) |
| Partner | 2.1 (9) |
| Can't remember | 0.5 (2) |
| **Mode of receiving the first FC if received from a provider (n = 326)** | |
| Offered by provider | 65.6 (214) |
| Requested by client | 30.9 (101) |
| Could not remember | 2.8 (9) |
| **Provider explained how to use first FC** | 98.4(420) |
| **Had enough information given for first FC use** | 74.9 (316) |
| **If offered choice what condom would you use** | |
| Choose the FC | 67.5 (288) |
| Choose the MC | 20.8 (89) |
| Like both FC/MC equally | 10.3 (44) |
| Don't like either MC/FC | 0.7 (3) |
| Not sure | 0.7 (3) |

*Question not asked of those who obtained from dispenser or other source.

South Africa's FC programme is the only example of a fully integrated national MC/FC condom programme globally that does not focus on a target risk group, and a rare example of a programme primarily funded by the government [11–13]. FC evaluations in other countries have focused on specific target groups or geographic locations [30–32], where unsuccessful programmes were found to have introduced the FC in an uncoordinated fashion without programmatic support [31]. A review of FC programmes globally has called for routine and intensive monitoring to inform outcome data better [7]. The South African programme has adopted this recommendation and commenced monitoring of FC distribution as part of the DHIS in 2013.

Despite availability, distribution levels of the FC relative to the MC are still low in South Africa, and this is reflective of FC uptake worldwide [4–5], and is attributed to higher cost, lack of male acceptance, and difficulties in use [4, 33–35]. The literature stresses that female-initiated male involvement is key for successful programming [33].

The results of this evaluation indicate that the condom programme has fulfilled all the recommended steps of the UNFPA 10-Step Strategic Approach to scale up Comprehensive Condom Programming (CCP) [36]. This ten-step guide includes *"the need to develop a comprehensive and integrated national strategy for male and female condoms".* The male and female condom programme are integrated in distribution mechanisms, both condoms are required to be available at all public-sector health facilities, and South Africa has a National Condom Distribution Plan [23]. Another key step is to monitor programme implementation and conduct research. All facilities have distribution targets and are required to report numbers of male and female condoms distributed. Evaluation findings presented here have been used to inform the National Department of Health in South Africa. Two of the steps are related to budget, including the development of a multi-year operational budget and mobilizing financial resources. Both of these steps have been completed and a budget has been set aside for condom procurement each year and a target number required for procurement; this is noted in the *HIV and AIDS and STI National Strategic Plan for South Africa, 2017–2022* [16]. Some of UNFPA's recommended steps could be strengthened in South Africa, including FC promotion, advocacy and engaging the media. The key recommendations for the SA and other FC programs include [37].

We suggest that the South African and other FC programmes include.

- Trained providers alone cannot be the only source of FC distribution—there is a need to make provision of FC supplies more accessible, particularly for experienced users.

- Although FCs are available widely in public health facilities, this level of health care is predominately utilised by women. Programmes need to ensure that FCs are more widely available to men.

- FC distribution should be closely monitored to ensure consistency of supplies and preclude stock-outs.

- Population-based national surveys should include the FC as a method disaggregated from the MC in order to monitor data on users' knowledge and use.

- Providers' negative attitudes and lack of technical counselling skills regarding the FC need to be addressed to improve uptake.

- The current branding of the FC is not appealing to men generally, especially those who have sex with men.

- Awareness-raising and marketing must be a priority to build demand.

## Conclusion

Findings from this evaluation provide solid support for further programme expansion in South Africa. As demand for FCs has increased, the NSP 2017-2022includes increased targets for condom distribution (3 billion MCs and 33 million FCs). Public-sector facilities have now been given distribution targets for both MCs and FCs (Personal communication, National Department of Health) and therefore this evaluation can be used to consider and address the realities of system, provider and client concerns. Years of limited distribution during the phased expansion of the programme may have conveyed to both providers and clients that FCs are not available at all sites, and that providers do not need to stock, promote and offer the product. The evaluation was conducted 20 years after the first sites had started distributing FCs and this was reflected in the range of years that each site had been participating in the programme. The final phase of expansion which occurred around 2012–2014 meant that some sites had been distributing for 2 years or less, whereas others had more r than 10 years of experience. Similarly, the phased introduction of new FC brands was evident during the FC evaluation, with some provinces not yet starting this component. Currently, all public-sector sites are distributing FCs. Sites with less experience in condom programming can learn from this evaluation. With the phased introduction of the donor-funded pre-exposure prophylaxis (PrEP) programme into South African health facilities, there are opportunities to learn and apply lessons learned from the evaluation of the national FC programme.

## Supporting information

**S1 File.**
(PDF)

**S2 File.**
(PDF)

**S3 File.**
(PDF)

**S4 File.**
(PDF)

**S5 File.**
(PDF)

## Acknowledgments

We thank the South African Department of Health-national, provincial, district and participating facilities for support and assistance with project planning and logistics. We also want to thank all participants for their time and contribution, and the non-public sector sites for their willingness to take part in our evaluation.

## Author Contributions

**Conceptualization:** Mags Beksinska, Zonke Mabude, Joanne E. Mantell, Cecilia Milford, Jennifer A. Smit.

**Data curation:** Mags Beksinska, Phumla Nkosi, Joanne E. Mantell.

**Formal analysis:** Mags Beksinska, Cecilia Milford.

**Funding acquisition:** Mags Beksinska, Joanne E. Mantell, Jennifer A. Smit.

**Investigation:** Mags Beksinska, Joanne E. Mantell, Cecilia Milford.

**Methodology:** Mags Beksinska, Phumla Nkosi, Zonke Mabude, Joanne E. Mantell, Bongiwe Zulu, Cecilia Milford, Jennifer A. Smit.

**Project administration:** Mags Beksinska, Phumla Nkosi, Zonke Mabude, Joanne E. Mantell, Bongiwe Zulu, Cecilia Milford, Jennifer A. Smit.

**Supervision:** Mags Beksinska, Phumla Nkosi, Zonke Mabude, Bongiwe Zulu, Jennifer A. Smit.

**Validation:** Mags Beksinska, Bongiwe Zulu.

**Writing – original draft:** Mags Beksinska, Joanne E. Mantell, Cecilia Milford, Jennifer A. Smit.

**Writing – review & editing:** Mags Beksinska, Phumla Nkosi, Zonke Mabude, Joanne E. Mantell, Bongiwe Zulu, Cecilia Milford, Jennifer A. Smit.

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
