## [Decision Letter · Decision Letter 0]

3 Dec 2019

PONE-D-19-26312

Global lessons from the evaluation of the South African National Female Condom Program

PLOS ONE

Dear Prof Beksinska,

Thank you for submitting your manuscript to PLOS ONE. After careful consideration, we feel that it has merit but does not fully meet PLOS ONE’s publication criteria as it currently stands. Therefore, we invite you to submit a revised version of the manuscript that addresses the points raised during the review process.

We would appreciate receiving your revised manuscript by Jan 17 2020 11:59PM. To enhance the reproducibility of your results, we recommend that if applicable you deposit your laboratory protocols in protocols.io, where a protocol can be assigned its own identifier (DOI) such that it can be cited independently in the future. For instructions see: http://journals.plos.org/plosone/s/submission-guidelines#loc-laboratory-protocols

We look forward to receiving your revised manuscript.

Kind regards,

Collins Iwuji, M.B;B.S, MSc, MD

Academic Editor

PLOS ONE

Journal Requirements:

1. Please include a copy of the interview guide used in the study, in both the original language and English, as Supporting Information, or include a citation if it has been published previously.

Reviewers' comments:

Reviewer's Responses to Questions

**Comments to the Author**

1. Is the manuscript technically sound, and do the data support the conclusions?

Reviewer #1: Partly

Reviewer #2: Partly

2. Has the statistical analysis been performed appropriately and rigorously? 

Reviewer #1: No

Reviewer #2: Yes

3. Have the authors made all data underlying the findings in their manuscript fully available?

Reviewer #1: No

Reviewer #2: Yes

4. Is the manuscript presented in an intelligible fashion and written in standard English?

Reviewer #1: No

Reviewer #2: Yes

5. Review Comments to the Author

Reviewer #1: Overall, the paper addresses and reports various aspects of the national female condom program in South Africa. The authors have presented comprehensive data and provided valuable insights from a range of government key stakeholders, health providers, and clients from seemingly representative samples from South Africa. However, the paper has many methodological weaknesses, incomplete presentation of the data, and lack of cohesive discussions and writing, which significantly weaken the quality of the manuscript.

Major comments:

1. In the beginning of the Materials and Methods section, I think there should be a section which provides the overview of the FC programs and guideline in South Africa as a background. Specifically, when the national FC program was launched in 2014 and expanded thereafter, were IEC and “demonstration models” supposed to be distributed to all clinic sites (if so, one model for FC demonstration per clinic)? Also, at the clinics, what were the specific guidelines/policies for healthcare providers (HCWs) in terms of FCs distribution to clinic clients? Were HCWs supposed to provide FCs to every client who visit the clinics or only to certain clients (i.e. attending for sexual and reproductive health issues, etc.)? Does the policy specify to provide FCs to younger clients (aged 15-19 years) as well? In my opinion, very insufficient information was provided by the authors such that it is difficult to properly assess the paper’s results.

2. Line 79-85: Although the authors mention that “the on-site assessment sample was selected randomly and proportionally” by the four criteria, the authors do not present any distribution of characteristics of the selected sites regarding these four criteria thus readers cannot assess how well this selection was performed and the overall characteristics of these sites. Please add this information in Table 1. Also, given that the data was collected nationally, it seems important to understand geographical distributions of the sites, at least at the provincial level.

3. Line 87-91: What was the rationale to choose tertiary education and social-marketing outlet for the surveys? Can the authors please elaborate what they mean by “social-marketing outlet”? I think authors should provide more explanation about distribution of FCs at tertiary education, etc., in the introduction (in line 64-67). Also, please provide more explanation about the list of non-public FC distributing sites such as the number of available non-public sites- were the sites only included if the sites were providing free FCs?

4. Line 101: What does it mean by “depending on total staff complement”? Recommend re-writing from 93 to 98 with more details.

5. Line 116-118: It is unclear what the authors mean by “the same three months…”. Does this mean that the data was collected for three months at each site? I think there should be a section under Methods to describe data collection at the sites in more details.

6. Table 1: can the authors break down and provide the type of providers (for provider interviews and, if possible, key informants) and the type of organizations included in non-public sector sites (the number of NGOs vs. tertiary education vs. …)? Also please provide information on sites (such as the number and variance) for the sub-sample besides the number of individual participants.

7. Line 133: Can the authors please elaborate in which languages the consents/interviews/self-administered questionnaires were offered? Given that this was done at the national level, I am not sure how many different languages were offered for questionnaires… and there is no explanation about translation of the languages in Methods.

9. Line 147-161: I think this information needs to be separately presented under “overview of national government policies and programs.”, and some of them (i.e. line 156-161) are more like a review of policy implementation process, rather than results. Also, how were these key informant interviews conducted? Were they semi-structured qualitative in-depth interviews? How were these interviews analysed (i.e. using what methods/analysis programs)? In Line 152, please specify and refer to the new FC brands.

10. Line 163-169: It’s really unclear to which data these results were referring. Are these based on the review of DHIS data? If so, please clearly state that.

11. Table 2: Please include p-values to make proper comparison between FC and MC. It’s very unclear whether the percentage in Table 2 presents the availability of FC or MC only OR the availability of any of condom, IEC materials, or demonstration model OR all of them. I strongly suggest to present results for availability of each of condom, IEC materials, and demonstration model separately in Table 2.

-Line 187: “higher” is not clear. What is the overall availability of the FCs vs. MCs?

-Line 188: please report the percentage of “FC leaflet availability”.

12. Line 209: Table 3: Please include p-values as the last column to compare between FCs vs. MCs.

13. Line 203: I think looking at the association among the different predictors is a completely different research question. To look at the predictors, I strongly believe that adjusted models such as logistic regression models adjusting for different predictors and characteristics need to be done. For line 204-206, if the reasons were asked in the surveys, please include them in Table 3. Also, please provide whether the provider interviews were done qualitatively or quantitatively (using survey forms) in the Methods. For the question ‘”In last month provider discussed…”, Being depended on the client vs. the frequency of providing the one-to-one sessions) are NOT mutually exclusive.

Furthermore, the question on “Demonstrate use to new users” seems confusing- was this referring to any time period or like in last month?

14. Line 212-217: Again, I strongly think that predictors related to provision of FCs need to be investigated in adjusted models including availability of condoms in every consultation room, being ever trained, types of providers, different provinces, experience of stock-outs, etc as potential predictors.

15. Line 223-228: I would like to see the complete Table for providers attitudes towards FCs and MCs, at least as a supplement, as well as p-value associated with that. Especially, I think this is really the key information given that the authors’ main conclusion that “providers are the gate-keepers” for FCs distribution.

16. Line 226-228: How were the question really asked to the respondents? I don’t think it would have been asked as “messy” or “weird” in the questionnaire… or was it? How was this concept asked in the survey? Or is it based on the qualitative interviews/findings? It it’s latter, I think very explicit methods need to be written regarding how this was conducted and analysed.

17. Line 237-242: How were these participants included? Was the self-administered survey offered to everyone who was visiting the clinic sites? What was the refusal vs. participation percentages? Also, given that this was done at the national level, I would like to see how these may differ by province, at least. Also, please include more demographic characteristics of clients (definitely, at least age groups, HIV status). Please include p-values for difference between women and men. Also, in the discussion, the authors discuss that “a high proportion of FC users were HIV-positive”… without any demographic characteristics presented in the results, it is not possible for readers to understand these results.

18. Line 250. Table 4. Please rephrase the title of the table, for example, “Clients’ knowledge and usage of FC in South Africa”.

19. The number of “Ever used FC” is obviously wrong (currently written as n=88 for women and n=587 for men), please correct this. Also, I also strongly recommend to report how many were current vs. previous FC users under the variable “Ever used FC”.

So.. out of 587+88 = 675 who ever used FCs, 427 (63%) completed the exit surveys? How were these people selected? Were they any demographic or systematic difference between those who completed the exist surveys vs. those who didn’t? What were the reasons for not completing the exit interviews?

20. Line 257-259: Please include this as part of Table 4 (i.e. Reasons for not using FCs among non FC-users* - then please give the detail description for this group as a footnote)

21. Line 271: Please carefully go through the table and see whether the percentage matches and adds up to 100%; for example, for the variable “First FC obtained from”, the percentages only add up to 95%- if other providers or sources were listed/selected, please list them as “others” category in the table. Also, I would rename “Offered FC or asked for first FC” as “Mode of receiving the first FC at clinics”.

22. Also please include the contents reported in 275-280 as part of Table 5 so the readers can understand fully.

23. Line 315-319: In the first few paragraphs of “Discussion” and throughout the Results section, the authors seem to point out gaps in the implementation and delivery of FC programs at national clinics then without much data supported, the authors claim that the evaluation seems to have fulfilled all the recommendation steps by the international organization. Could the authors please elaborate how the SA program has fulfilled the recommended steps in more details? Also please add an overall conclusion section at the end of line 331.

Minor comments:

- I think the word “global” lesson can be misleading as the study presents the data from South Africa alone. I suggest to remove the word “global” in this case.

- Line 95: what does IEC stand for? Information, Education, and Communication? Please state at least once before using the acronym.

- Line 284: Please change “Conclusion” to “Discussion”.

- Please check the references for consistency and correct formats.

- There are many grammatical errors and incomplete sentences. The paper needs to be proofread. For example,

Line 49: put -s after “sexually transmitted infection” and introduce acronym here, and thereafter use STI (for example, in Line 79)

Line 57: please spell out WHO/UNFPA when used for the first time

Line 57: need a reference for “others in development”.

Line 67: please put a reference.

Line 79: please replace to “the national STIs sentinel surveillance sites”

Line 80: includes -> include

Line 95: sites/stock outs/expired stock/sub-distribution to other sites; please do not use “/”,

Line 112: Remove “KIs” – I don’t think the authors used this acronym after this

Line 124: detail -> details

Line 110: ever heard of -> being ever heard of or using FC; the sentences need to be re-written.

Line 134: Results and � Results

Line 135: Please provide months when the data collection started (in 2014) and ended in 2016.

Line 151: Please remove – after “with”

Table 2: remove “day of telephonic survey” ; remove “condoms” after “Distribution points”; Please put the list of acronym as footnotes under the table. Please put the label as %(n) in the second row.

Line 200: and -> or

Line 304: Please rephase “… a rare example of one funded primarily by the National Government”- to, for example, “a rare example of the program primarily funded by the government.” Please add references.

Line 309-10. “Notable… in 2013” seems out of context or provide insufficient details.

Line 311-313: Difficult to read and awkwardly written. Please revise.

Throughout the paper, the authors keep using the word “variable”. I would recommend to diversity the term.

Reviewer #2: This is an interesting paper evaluating one of the only national female condom provision programmes globally. As such it has the potential to give important programmatic insights. However, there are issues with the way the methods are described, and the current structure of the results means there is a lack of clarity in the findings. This paper requires substantial revision prior to being considered for publication.

1. INTRODUCTION: a short paragraph introducing FC more broadly (efficacy, effectiveness, global uptake, acceptability) would provide important context.

2. METHODS:

i. This is described as a mixed methods evaluation comprising surveys, interviews and on site assessments. However, it appears that the interviews are structured (rather than qualitative interviews). It is important to be aware that mixed methods research pertains to the combination of QUANTITATIVE AND QUALITATIVE data in a study.

ii. Could the authors clarify exactly what types of data (quant and qual) they collected?

iii. If it is just quantitative data collected then this is NOT a mixed methods evaluation and the methods need to be amended to reflect this.

iv. If the interviews (key informant, provider and client) are qualitative then please (a) describe these components in detail (b) give a justification for why a mixed methods approach was used (c) describe the mixed methods model you are using (i.e. convergent parallel design) and (d) discuss how quant and qual data were integrated (which needs to be done in mixed methods analysis) . I suggest the authors look at O’Cathain, A., Murphy, E., and Nicholl, J. (2008) ‘The quality of mixed methods studies in health services research’, Journalof Health Services Research and Policy, vol. 13, no. 2, pp. 92-98.

v. Data sources are currently unclear - I suggest listing each sample (e.g. nationwide sites), sub-samples of sites etc (and then for each listign exactly what data collection methods were used). How were key informants selected?

vi. More detail is required on: surveys (what was asked, standardised, any validated tools e.g. attitudinal questions), interviews (were they quantitative or qualitative), on site assessments (what did this entail, who conducted them)., how were surveys administered, who conducted the interviews

vii. More details on DHIS sata source

3. Data analysis: if qualitative data were analysed how were these analysed, how were qual and quant data integrated, more details on stats (key variables, key outcomes of interest, you present p values - state you are using chi squared tests - wondering why you did not consider multivariable logistic models)

4. Line 134: incomplete subheading

5. Results need to be re structured for clarity as currenlty hard to know what are sources of data - start with broad overview from DHIS and national telephone survey, then KI and provider, and then client. If you have qualitative data , consider use of quotes to support your results.

6. Tables: present p-values for comparisons please

7. You cannot assess predictors as this is a cross sectional design - you are looking at associations.

8. I am struck by the fact that 15% of providers did nOT know that FC were not reusable - these findings need to be brought out more.

9. Line 240 fewer rather than less

10.Line 259 "where"

11. I was unsure what the exit interviews were

12. Conclusions: strengths and weaknesses of this evaluation, recommendations for further work, highlight the important gaps in knowledge and attitudes towards FC - how can this be addressed.

6. PLOS authors have the option to publish the peer review history of their article (what does this mean?). If published, this will include your full peer review and any attached files.

Reviewer #1: No

Reviewer #2: Yes: Shema Tariq

---

## [Author Response · Author response to Decision Letter 0]

1 Jun 2020

Editor

PloS One

We appreciate the thoughtful comments of the two reviewers and below we respond by-by-point in bold font to each comment or query. Reviewers’ comments were also a catalyst for substantial editing and reorganization. We have uploaded both highlighted and clean versions of the manuscript. Reference to lines refer to those in the clean version of the manuscript.

1. Please include a copy of the interview guide used in the study, in both the original language and English, as Supporting Information, or include a citation if it has been published previously.

We have included a copy of our Interview Guide for Key Informants as Supplementary Material. This guide has not been published.

We have deleted reference to “data not shown” from the text. We have added a figure showing the data. 

Reviewers' comments:

Reviewer's Responses to Questions

Comments to the Author

1. Is the manuscript technically sound, and do the data support the conclusions?

Reviewer #1: Partly

Reviewer #2: Partly

2. Has the statistical analysis been performed appropriately and rigorously? 

Reviewer #1: No

Reviewer #2: Yes

3. Have the authors made all data underlying the findings in their manuscript fully available?

Reviewer #1: No

Reviewer #2: Yes

4. Is the manuscript presented in an intelligible fashion and written in standard English?

Reviewer #1: No

Reviewer #2: Yes

5. Review Comments to the Author

Authors comment: We have been asked to add a lot of information and the article has double din length and so have the number of tables we had so many changes its was almost impossible to view in tack changes and so we have highlighted the additional sections in yellow.

Reviewer #1: Overall, the paper addresses and reports various aspects of the national female condom program in South Africa. The authors have presented comprehensive data and provided valuable insights from a range of government key stakeholders, health providers, and clients from seemingly representative samples from South Africa. However, the paper has many methodological weaknesses, incomplete presentation of the data, and lack of cohesive discussions and writing, which significantly weaken the quality of the manuscript.

Major comments

1. In the beginning of the Materials and Methods section, I think there should be a section which provides the overview of the FC programs and guideline in South Africa as a background. Specifically, when the national FC program was launched in 2014 and expanded thereafter, were IEC and “demonstration models” supposed to be distributed to all clinic sites (if so, one model for FC demonstration per clinic)? Also, at the clinics, what were the specific guidelines/policies for healthcare providers (HCWs) in terms of FCs distribution to clinic clients? Were HCWs supposed to provide FCs to every client who visit the clinics or only to certain clients (i.e. attending for sexual and reproductive health issues, etc.)? Does the policy specify to provide FCs to younger clients (aged 15-19 years) as well? In my opinion, very insufficient information was provided by the authors such that it is difficult to properly assess the paper’s results.

Given the vast amount of data to synthesize, we were challenged with how much information and data to include in this manuscript. We have now added additional back ground information at the end of the Introduction (Lines 70-95) rather than the beginning of the methods as suggested, as this section starts with a discussion of the FC globally and then provides FC information about South Africa specifically. There were no written guidelines regarding a focus on a particular type of client or key population. The FC pilot programme commenced in 1998, not in 2014, and was phased into a full-scale national programme in 2014. We have updated the history of the FC programme to improve clarity of its development.

2. Lines 79-85: Although the authors mention that “the on-site assessment sample was selected randomly and proportionally” by the four criteria, the authors do not present any distribution of characteristics of the selected sites regarding these four criteria thus readers cannot assess how well this selection was performed and the overall characteristics of these sites. Please add this information in Table 1. Also, given that the data was collected nationally, it seems important to understand geographical distributions of the sites, at least at the provincial level.

We have revised Table 1 (which is a new table) to include an overview of the data collection methods, whether all sites or sub-set of sites, key constructs presented in this paper and target population. We developed new tables (Table 2 & 3) to present characteristics of the sites by province. 

3. Lines 87-91: What was the rationale to choose tertiary education and social-marketing outlet for the surveys? Can the authors please elaborate what they mean by “social-marketing outlet”? I think authors should provide more explanation about distribution of FCs at tertiary education, etc., in the introduction (in Lines 64-67). Also, please provide more explanation about the list of non-public FC distributing sites such as the number of available non-public sites- were the sites only included if the sites were providing free FCs?

We selected a tertiary education site because all tertiary education sites in South Africa are targeted for the FC (and MC) programme. The government of South Africa also launched socially marketed FC (managed by the local affiliate of Population Services International [PSI]), selling FCs branded under the same name as socially marketed male condoms known as “Lovers Plus”. A “Social-marketing outlet” in our study was defined as a retail outlet (store/petrol station etc) that sold female condoms. We added more information about both tertiary institutions and the social marketing introduction in the Introduction Section (Lines 88-96) and also in the Selection of Sites Section (Lines 109 143). All FCs are provided free in South Africa (public sector, NGOs, tertiary institutions). The private sector market rarely stocks FCs.

4. Line 101: What does it mean by “depending on total staff complement”? Recommend re-writing from 93 to 98 with more details.

Total staff complement refers to the total number of staff at the facility. We have edited this sentence and added more detail on this. We interviewed two staff when total staff employed were less than five and three staff when the total staff employed were five or more. (Lines 198-199)

5. Lines 116-118: It is unclear what the authors mean by “the same three months…”. Does this mean that the data was collected for three months at each site? I think there should be a section under Methods to describe data collection at the sites in more details.

We have added more details about data collection methods in the Methods Section. We reviewed the DHIS data for the same three months (February through April 2014). (Lines 173-174)

6. Table 1: can the authors break down and provide the type of providers (for provider interviews and, if possible, key informants) and the type of organizations included in non-public sector sites (the number of NGOs vs. tertiary education vs. …)? Also please provide information on sites (such as the number and variance) for the sub-sample besides the number of individual participants.

The majority of providers were nurses (73%) as stated in the text; the rest comprised one or more of 13 other types of positions and we have included all categories in new Table 5. We have added Key informants into Table 2. We have, as above, provided more detail on sites by province but not individual site as there are too many of them. 

7. Line 133: Can the authors please elaborate in which languages the consents/interviews/self-administered questionnaires were offered? Given that this was done at the national level, I am not sure how many different languages were offered for questionnaires… and there is no explanation about translation of the languages in Methods.

The client exit interview and client anonymous survey were offered to participants in all of South Africa’s 11 languages; similarly, the consent for the client exit interview was offered in all 11 languages. This is noted in the Methods Section. (Lines 146-148)

9. Lines 147-161: I think this information needs to be separately presented under “overview of national government policies and programs.”, and some of them (i.e. line 156-161) are more like a review of policy implementation process, rather than results. Also, how were these key informant interviews conducted? Were they semi-structured qualitative in-depth interviews? How were these interviews analysed (i.e. using what methods/analysis programs)? In Line 152, please specify and refer to the new FC brands.

We have presented this information in a new section, Overview of national government policies and programmes. (Lines 288-346)

We have added more information about how these interviews were conducted in the Methods Section. (Lines 155-167)

Analysis of these interviews is described in the Data Analysis Section. (Lines 224-231)

We have mentioned the new FC brands in the Introduction Section. (Lines 79-80) 

10. Line s163-169: It’s really unclear to which data these results were referring. Are these based on the review of DHIS data? If so, please clearly state that.

Lines 163-169 do not refer to the DHIS. The DHIS data were only used to verify the distribution figures given for the 3-month site distribution to see if they matched what the clinic had documented in their distribution logs. We have added a sub-header for this section starts that we are talking about data from the site assessment to make it clearer. (Lines 169-174)

11. Table 2: Please include p-values to make proper comparison between FC and MC. It’s very unclear whether the percentage in Table 2 presents the availability of FC or MC only OR the availability of any of condom, IEC materials, or demonstration model OR all of them. I strongly suggest to present results for availability of each of condom, IEC materials, and demonstration model separately in Table 2.

We have presented the condom availability, IEC and demonstration models separately for the male and female condom in Table 4. 

The lower part of the table was inadvertently deleted. We have added the p-values to the table. 

12. Line 187: “higher” is not clear. What is the overall availability of the FCs vs. MCs?

Table 4 presents the proportion of each type of distribution point in each facility where condoms are available to clients. We have edited the text to explain that higher means available to clients at that distribution point. (line 378)

-Line 188: please report the percentage of “FC leaflet availability”. We have added FC leaflet availability in Table 4. 

12. Line 209: Table 3: Please include p-values as the last column to compare between FCs vs. MCs.

We added p values in new Table 4 (previously table 3). 

13. Line 203: I think looking at the association among the different predictors is a completely different research question. To look at the predictors, I strongly believe that adjusted models such as logistic regression models adjusting for different predictors and characteristics need to be done. For line 204-206, if the reasons were asked in the surveys, please include them in Table 3. Also, please provide whether the provider interviews were done qualitatively or quantitatively (using survey forms) in the Methods. For the question ‘”In last month provider discussed…”, Being depended on the client vs. the frequency of providing the one-to-one sessions) are NOT mutually exclusive.

Furthermore, the question on “Demonstrate use to new users” seems confusing- was this referring to any time period or like in last month?

There are a number of point asked in the above point. We have answered them in order here:-We believe that descriptive data are sufficient for this evaluation and that its merit is not diminished by the lack of logistic regression analyses. The protocol did not specify conducting logistic regression and the sample size was not powered to do so. At the time of receiving the sample we were not aware of the number of providers in those facilities and we were only in each facility for a short time and often many did not have time for an interview. Therefore providers were purposively sampled.

We are not sure what was meant by the reasons in table 3 (now table 4) in the reviewers question above. The exact options for each question is in the table? 

Demonstrate to new users means users who have never used an FC before

We have added the responses (clients can read instructions, no time in consultation and no demonstration model) to Table 6 (previously table 4).

The provider survey was quantitative; we have noted this in the Methods Section. (Line 197)

The response to “discussed in last month” was often responded to by the provider as “depended on client”; this is a common response as providers often do not want to discuss condom use with particular types of clients, e.g., folder or married women. This was a verbatim response was reported by nearly a quarter of providers regarding the female condom and nearly a fifth with regard to the male condom. We therefore consider this to be a valid response and have opted not to exclude this response category. We have added an explanatory footnote at the bottom of this table. (now Table 6)

The question asking about demonstration to new users was not asked within a time frame (last month, etc). We assumed that the providers’ response would refer to what they usually do in terms of demonstrating FC use or not to new users. Other questions specified a window period, including asking about discussing condom use in last month and for having personally given condoms to a male or female client in last week.

14. Line 212-217: Again, I strongly think that predictors related to provision of FCs need to be investigated in adjusted models including availability of condoms in every consultation room, being ever trained, types of providers, different provinces, experience of stock-outs, etc as potential predictors.

This type of analysis was not described in the protocol and we believe it the present descriptive analyses have merit. It would be complex as there is a variety of number and types of consultation rooms. Some consultation rooms were limited in space and so condoms had to be kept in reception or in toilets or with security guards. The type of provider, aside from nurses, included about 20 different categories many with only one in each and so it would not be possible to do this type of analysis (pharmacy assistant, project manager, volunteer, condom champion etc). Also staffing differs along the line of superiority and having an administrative load, resulting in some staff attending to clients for longer periods than others. 

There was almost no stock-outs (2.8%) of sites and when they occurred they were often for a range of reasons which are now described in the Results Section (Lines 351-354). Our sample size was adequate at national level but we feel its inadequate at provincial level (9 provinces) as each province was so different we do not want to make too many speculations between provinces. The provinces are so different, population-wise, geographically (urban vs. rural). We were trying to give a national overview in this paper. Finally, with the different variables and target populations across data collection, methods, adjusted analyses and a meta-analysis seem less important than the analyses we have presented in this manuscript. 

15. Line 223-228: I would like to see the complete Table for providers attitudes towards FCs and MCs, at least as a supplement, as well as p-value associated with that. Especially, I think this is really the key information given that the authors’ main conclusion that “providers are the gate-keepers” for FCs distribution.

We have added a new table (Table 7) to the main text. We have not looked at the association of these items with ever trained in FC as many did not know the information as they had not been trained

16. Line 226-228: How were the question really asked to the respondents? I don’t think it would have been asked as “messy” or “weird” in the questionnaire… or was it? How was this concept asked in the survey? Or is it based on the qualitative interviews/findings? It it’s latter, I think very explicit methods need to be written regarding how this was conducted and analysed.

The items are ‘female condoms are messy’ and ‘female condoms are weird’. These items were derived from Neilands and Choi (2002) and have been used in other studies in sub-Saharan Africa, although not with providers. Reference is now provided.

17. Line 237-242: How were these participants included? Was the self-administered survey offered to everyone who was visiting the clinic sites? What was the refusal vs. participation percentages? Also, given that this was done at the national level, I would like to see how these may differ by province, at least. Also, please include more demographic characteristics of clients (definitely, at least age groups, HIV status). Please include p-values for difference between women and men. Also, in the discussion, the authors discuss that “a high proportion of FC users were HIV-positive”… without any demographic characteristics presented in the results, it is not possible for readers to understand these results.

The survey was offered to everyone in the waiting area in the morning of the visit. Staff left them at the reception, left them on tables and chairs. People did not need to refuse and so we do not have these data. The facilities we visited were often very busy and it would not have been possible to offer individually (Lines 201-209). HIV status was not collected in this anonymous client survey as we felt it may dissuade participation. The questionnaire focused on FC use. We collected data only on age and gender (Lines 510 onwards). Table 8 (previously table 4) presents survey data by participant gender. The purpose of the survey was to look at uptake. We also have added tabled showing the distribution of data collection method by province and another table displaying site location, facility type, years of FC distribution (Table 3), and more than 1 type of FC available at site for each province (Table 3). 

18. Line 250. Table 4. Please rephrase the title of the table, for example, “Clients’ knowledge and usage of FC in South Africa”.

We have renamed the title of Table 8 (previously table 4) as suggested.

19. The number of “Ever used FC” is obviously wrong (currently written as n=88 for women and n=587 for men), please correct this. Also, I also strongly recommend to report how many were current vs. previous FC users under the variable “Ever used FC”.

We agree and realize that we had inadvertently entered incorrect numbers for men and women. This has been corrected in Table 8 (previously table 4). The question in the survey asked about use with current partner. We have added this information to the table.

So.. out of 587+88 = 675 who ever used FCs, 427 (63%) completed the exit surveys? How were these people selected? Were they any demographic or systematic difference between those who completed the exist surveys vs. those who didn’t? What were the reasons for not completing the exit interviews?

The client exit survey was not linked to the client anonymous survey. The populations for the two surveys were different. Therefore, we cannot say that 63% of those who completed the exit survey had ever used the DC. During the facility visit, the research staff informed clients during the day in the different waiting areas that any ever or current users could volunteer for an interview at the end of their consultation. We have added this information in the Client exit interviews section (Lines 211-219). The exit interviews took almost an hour (line 216) and meant that clients would have had to stay after their consultation; this may not have been possible for all clients. 

The anonymous survey only took a minute or two to complete. Since it was anonymous, we did not want to target clients based on demographic or condom use characteristics. In fact, we wanted to identify how many clients had ever or never used FCs. Although some of the anonymous survey participants also completed the exit interview, we did not record if they had or not so as linkage would have violated the anonymity of the survey (Lines 201-202). We did not collect data on the number of clients who refused participation in the anonymous client survey or client exit survey. The female clients who participated in the exit interviews were women who self-selected to be interviewed.

20. Line 257-259: Please include this as part of Table 4 (i.e. Reasons for not using FCs among non FC-users* - then please give the detail description for this group as a footnote)

We have added the reasons for non-use in a new Figure 2 (Line 543), but we do not have a detailed description of these participants as only age and sex data were collected. We would not have been able to collect detailed on all 4,000+ clients. We made the anonymous survey was designed to be brief and simple as possible.

21. Line 271: Please carefully go through the table and see whether the percentage matches and adds up to 100%; for example, for the variable “First FC obtained from”, the percentages only add up to 95%- if other providers or sources were listed/selected, please list them as “others” category in the table. Also, I would rename “Offered FC or asked for first FC” as “Mode of receiving the first FC at clinics”.

Percentages do not necessarily add up to 100% in Table 9 if multiple responses are allowed (e.g., main reasons for first FC use). Percentages for source of obtaining first FC and mode of receiving the first FC, and if offered choice of condom total 100%. In Table 9 we have renamed the Offered FC to Mode of receiving FC.

22. Also please include the contents reported in 275-280 as part of Table 5 so the readers can understand fully.

We have added some of the information in the text to Table (previously table 5) but we did not want to repeat it all in the text and table as most of the text information was simply a yes or no answer.

23. Line 315-319: In the first few paragraphs of “Discussion” and throughout the Results section, the authors seem to point out gaps in the implementation and delivery of FC programs at national clinics then without much data supported, the authors claim that the evaluation seems to have fulfilled all the recommendation steps by the international organization. Could the authors please elaborate how the SA program has fulfilled the recommended steps in more details? Also please add an overall conclusion section at the end of line 331.

We have added the information into the discussion as suggested. We have mentioned the required steps and how the program has fulfilled them (Lines 611-622). 

Minor comments:

- I think the word “global” lesson can be misleading as the study presents the data from South Africa alone. I suggest to remove the word “global” in this case.

We have removed the word ‘global’.

- Line 95: what does IEC stand for? Information, Education, and Communication? Please state at least once before using the acronym.

We have written out IEC (Line 182)

.

- Line 284: Please change “Conclusion” to “Discussion”. 

We have changed the Conclusion header to Discussion header .

- Please check the references for consistency and correct formats.

We have checked the references for consistency of format and corrected them.

- There are many grammatical errors and incomplete sentences. The paper needs to be proofread. For example,

Line 49: put -s after “sexually transmitted infection” and introduce acronym here, and thereafter use STI (for example, in Line 79) done

Line 57: please spell out WHO/UNFPA when used for the first time done 

Line 57: need a reference for “others in development”. added

Line 67: please put a reference. added

Line 79: please replace to “the national STIs sentinel surveillance sites” done

Line 80: includes -> include done

Line 95: sites/stock outs/expired stock/sub-distribution to other sites; please do not use “/”, removed

Line 112: Remove “KIs” – I don’t think the authors used this acronym after this. We have retained ‘KI’ as we have used this term more than once. 

Line 124: detail -> details done

Line 110: ever heard of -> being ever heard of or using FC; the sentences need to be re-written.

Line 134: Results and � Results We do not understand what the Reviewer is suggesting. 

Line 135: Please provide months when the data collection started (in 2014) and ended in 2016. Done

Line 151: Please remove – after “with” done

Table 2: remove “day of telephonic survey”; done remove “condoms” after “Distribution points”; done Please put the list of acronym as footnotes under the table these are now in text. Please put the label as % (n) in the second row. done

Line 200: and -> or done

Line 304: Please rephase “… a rare example of one funded primarily by the National Government”- to, for example, “a rare example of the program primarily funded by the government.” Please add references. Done and added reference

Line 309-10. “Notable… in 2013” seems out of context or provide insufficient details. done

Line 311-313: Difficult to read and awkwardly written. Please revise. reworded

Throughout the paper, the authors keep using the word “variable”. I would recommend to diversity the term. revised

Reviewer #2: This is an interesting paper evaluating one of the only national female condom provision programmes globally. As such it has the potential to give important programmatic insights. However, there are issues with the way the methods are described, and the current structure of the results means there is a lack of clarity in the findings. This paper requires substantial revision prior to being considered for publication.

1. INTRODUCTION: a short paragraph introducing FC more broadly (efficacy, effectiveness, global uptake, acceptability) would provide important context.

We have added more information about the FC in the Introduction Section. (Lines 51-60; 71-96)

2. METHODS:

i. This is described as a mixed methods evaluation comprising surveys, interviews and on site assessments. However, it appears that the interviews are structured (rather than qualitative interviews). It is important to be aware that mixed methods research pertains to the combination of QUANTITATIVE AND QUALITATIVE data in a study.

The Key Informant interviews were qualitative. We have noted this in the Methods Section and in Table 1. In our revisions, we added more information form the key informant interviews (lines 254-167) and integrated them in sections that also report quantitative data. 

ii. Could the authors clarify exactly what types of data (quant and qual) they collected?

We have added more information in the Methods Section. The key informant interview was the only qualitative tool. The other tools were quantitative.

iii. If it is just quantitative data collected then this is NOT a mixed methods evaluation and the methods need to be amended to reflect this.

Please see response in As noted above in 2i and 2ii.

iv. If the interviews (key informant, provider and client) are qualitative then please (a) describe these components in detail (b) give a justification for why a mixed methods approach was used (c) describe the mixed methods model you are using (i.e. convergent parallel design) and (d) discuss how quant and qual data were integrated (which needs to be done in mixed methods analysis) . I suggest the authors look at O’Cathain, A., Murphy, E., and Nicholl, J. (2008) ‘The quality of mixed methods studies in health services research’, Journal of Health Services Research and Policy, vol. 13, no. 2, pp. 92-98.

We have added more detail about all data collection methods and a new table (Table 1) . As we stated above, the evaluation used a mixed-methods approach. All of the data collection used quantitative tools, with the exception of the Key Informant Interview tool. The mixed-methods evaluation used a convergent concurrent or parallel) design. We used a mixed-methods approach, albeit limited qualitative data collection, to elicit a better understanding of the context and enrich perspectives of system-level FC issues (Lines 151-152) (which is better served by use of qualitative methods). We limited use of qualitative methods because of the need for timely completion of the evaluation. 

v. Data sources are currently unclear - I suggest listing each sample (e.g. nationwide sites), sub-samples of sites etc (and then for each listing exactly what data collection methods were used). How were key informants selected?

vi. More detail is required on: surveys (what was asked, standardised, any validated tools e.g. attitudinal questions), interviews (were they quantitative or qualitative), on site assessments (what did this entail, who conducted them)., how were surveys administered, who conducted the interviews

vii. More details on DHIS data source 

We have added more detail on all data collection methods in the text (Lines 127 onwards) and in a new table, Table 1. We note that all data collection was conducted by a cadre of research interviewers trained in quantitative and qualitative methods . Information on DHIS has been expanded (Lines 169-173). 

3. Data analysis: if qualitative data were analysed how were these analysed, how were qual and quant data integrated, more details on stats (key variables, key outcomes of interest, you present p values - state you are using chi squared tests - wondering why you did not consider multivariable logistic models)

See response to Reviewer 1 comment #14

4. Line 134: incomplete subheading

This incomplete heading has been corrected.

5. Results need to be re structured for clarity as currently hard to know what are sources of data - start with broad overview from DHIS and national telephone survey, then KI and provider, and then client. If you have qualitative data, consider use of quotes to support your results.

We have made the sources of data clearer with sub-headings and started with the broader overview. Key informant data were discussed first in the Results section because the data are overarching and not province- or site-specific. 

6. Tables: present p-values for comparisons please

We have added p-values for comparisons in Tables 4 and 8. 

7. You cannot assess predictors as this is a cross sectional design - you are looking at associations. 

We agree that any bivariate analyses look at associations of variables and have deleted reference to predictors.

8. I am struck by the fact that 15% of providers did nOT know that FC were not reusable - these findings need to be brought out more.

Research on the first polyurethane FC (FC1) that indicated it could be reused several times after washing and drying. However, as the material in FC2 changed from polyurethane to synthetic latex, this was no longer possible. We have added this to the Discussion Section (Line 577 583 ).

9. Line 240 fewer rather than less changed

10.Line 259 "where" changed

11. I was unsure what the exit interviews were

Exit interviews were conducted after clients had completed their visit to the clinic. We have explained this (Lines 206-213).

12. Conclusions: strengths and weaknesses of this evaluation, recommendations for further work, highlight the important gaps in knowledge and attitudes towards FC - how can this be addressed.

6. PLOS authors have the option to publish the peer review history of their article (what does this mean?). If published, this will include your full peer review and any attached files.

Do you want your identity to be public for this peer review? For information about this choice, including consent withdrawal, please see our Privacy Policy.

Reviewer #1: No

Reviewer #2: Yes: Shema Tariq

---

## [Decision Letter · Decision Letter 1]

20 Jul 2020

LESSONS FROM THE EVALUATON OF THE SOUTH AFRICAN NATIONAL  FEMALE CONDOM PROGRAMME

PONE-D-19-26312R1

Dear Dr. Beksinska,

We’re pleased to inform you that your manuscript has been judged scientifically suitable for publication and will be formally accepted for publication once it meets all outstanding technical requirements.

Kind regards,

Collins Iwuji, M.B;B.S, MSc, MD

Academic Editor

PLOS ONE

Additional Editor Comments (optional):

Title of Table 1 is missing. Please include.

Reviewers' comments:

Reviewer's Responses to Questions

**Comments to the Author**

1. If the authors have adequately addressed your comments raised in a previous round of review and you feel that this manuscript is now acceptable for publication, you may indicate that here to bypass the “Comments to the Author” section, enter your conflict of interest statement in the “Confidential to Editor” section, and submit your "Accept" recommendation.

Reviewer #1: All comments have been addressed

2. Is the manuscript technically sound, and do the data support the conclusions?

Reviewer #1: Yes

3. Has the statistical analysis been performed appropriately and rigorously? 

Reviewer #1: Yes

4. Have the authors made all data underlying the findings in their manuscript fully available?

Reviewer #1: Yes

5. Is the manuscript presented in an intelligible fashion and written in standard English?

Reviewer #1: Yes

6. Review Comments to the Author

Reviewer #1: Thank you for the opportunity to re-review the manuscript by Beksinska et al. The authors have well addressed my comments and strengthened the manuscript significantly by including more details about the methodology and study results. Overall, I appreciate clarity by including more detailed information but some of the information presented in the main text can be considered for the supplement. Especially, with clarify in the methods for collecting different data sources, I think the readers can now fully appreciate the comprehensive works done by the authors. So I'd recommend the results section and tables in the main text focus on the comprehensive findings across different data sources from the study and include any additional tables (for example, Table 2 and 3) presented in the supplement as per the authors' and the editor's judgement.

Minor comments:

1. Heading for Table 1 seems to be missing on page 7.

2. Table 2 and 3 are very helpful for the readers to understand the characteristics of selected sites but if the number of Tables need to be reduced, I believe these can be presented in supplement and briefly summarized in the main text (as the authors presented but perhaps even in a more summarized fashion).

3. I think the quote in line 304-310 can be deleted as the authors well summarize this in line 301-302. Also, I am not sure much detail regarding the NDoH’s policy reviews are needed to be included in the line 299-318 under the Results section.

4. For the quotes from the key informant interviews, it could be more informative if the acronym can be explained a little bit more. For example, “FC-NGO-KI-GP01” can be “A general practitioner in NGO, KI” or “DIS KZN01” can be “a district manager, KZN”.

5. Table 5 is helpful but can be grouped into a fewer number of categories for meaningful difference among job roles. Also, this can be included as a supplement if needed.

6. Likewise, Table 7 is helpful to understand the results but given the length of the questionnaire, I wonder the results need to be included as a supplement or perhaps sub-grouped or presented in a simpler format? I defer that decision to the authors and the editor.

7. PLOS authors have the option to publish the peer review history of their article (what does this mean?). If published, this will include your full peer review and any attached files.

Reviewer #1: No

---

## [Editor Report · Acceptance letter]

29 Jul 2020

PONE-D-19-26312R1 

LESSONS FROM THE EVALUATON OF THE SOUTH AFRICAN NATIONAL  FEMALE CONDOM PROGRAMME 

Dear Dr. Beksinska:

I'm pleased to inform you that your manuscript has been deemed suitable for publication in PLOS ONE. Congratulations! Your manuscript is now with our production department. 

Kind regards, 

on behalf of

Dr. Collins Iwuji 

Academic Editor

PLOS ONE